# Double Filtration Plasmapheresis with Polyvinyl Alcohol-Based Membrane Lowers Serum Inflammation and Toxins in Patients with Hyperlipidemia

**DOI:** 10.3390/bioengineering10010089

**Published:** 2023-01-09

**Authors:** Wen-Sheng Liu, Chien-Hung Lin, Ching-Yao Tsai, Hsiang-Tsui Wang, Szu-Yuan Li, Tsung-Yun Liu, Ann Charis Tan, Han-Hsing Tsou, Kuo-Hsien Tseng, Chih-Ching Lin

**Affiliations:** 1Division of Nephrology, Department of Medicine, Taipei City Hospital, Zhongxing Branch, Taipei 103, Taiwan; 2School of Medicine, National Yang Ming Chiao Tung University, Hsinchu 300, Taiwan; 3College of Science and Engineering, Fu Jen Catholic University, New Taipei City 242, Taiwan; 4Institute of Food Safety and Health Risk Assessment, School of Pharmaceutical Sciences, National Yang Ming Chiao Tung University, Hsinchu 300, Taiwan; 5Department of Special Education, University of Taipei, Taipei 100, Taiwan; 6Department of Pediatrics, Taipei Veterans General Hospital, Taipei 112, Taiwan; 7Institute of Public Health, Department of Public Health, College of Medicine, National Yang Ming Chiao Tung University, Hsinchu 300, Taiwan; 8Department of Ophthalmology, Taipei City Hospital, Zhongxing Branch, Taipei 103, Taiwan; 9Department of Business Administration, Fu Jen Catholic University, New Taipei City 242, Taiwan; 10Department of Pharmacology, National Yang Ming Chiao Tung University, Hsinchu 300, Taiwan; 11Division of Nephrology, Department of Medicine, Taipei Veterans General Hospital, Taipei 112, Taiwan; 12Kim Forest Enterprise Co., Ltd., New Taipei City 221, Taiwan; 13Division of Nephrology, Department of Medicine, Taipei Veterans General Hospital, Taoyuan Branch, Taoyuan 330, Taiwan

**Keywords:** hyperlipidemia, cardiovascular disease (CVD), double filtration (DF), inflammation, perfluorochemicals (PFC)

## Abstract

Hyperlipidemia is increasing in prevalence and is highly correlated with cardiovascular disease (CVD). Lipid-lowering medications prevent CVD but may not be suitable when the side effects are intolerable or hypercholesterolemia is too severe. Double-filtration plasmapheresis (DF) has shown its therapeutic effect on hyperlipidemia, but its side effects are not yet known. We enrolled 45 adults with hyperlipidemia in our study. The sera before and two weeks after DF were evaluated, and we also analyzed perfluorochemicals to see if DF could remove these lipophilic toxins. After DF, all lipid profile components (total cholesterol, triglycerides, high-density lipoprotein [HDL], and low-density lipoprotein [LDL]) had significantly decreased. Leukocyte counts increased while platelet levels decreased, which may have been caused by the puncture wound from DF and consumption of platelets during the process. As for uremic toxins and inflammation, levels of C-reactive protein, uric acid, and alanine transaminase (ALT) all decreased, which may be related to the removal of serum perfluorooctane sulfonate (PFOS) and improvement of renal function. The total cholesterol/HDL ratio and triglycerides were significantly higher in the diabetes mellitus (DM) group at baseline but did not significantly differ after DF. In conclusion, DF showed potential for improving inflammation and removing serum lipids and PFOS in adults with hyperlipidemia.

## 1. Introduction

Hyperlipidemia’s prevalence is increasing in response to changing dietary habits as part of the modern lifestyle. This rising prevalence has correlated with higher rates of cardiovascular disease (CVD) and stroke. [1]

Many lipid-lowering agents have shown a therapeutic effect on hyperlipidemia. However, medication’s effects in certain populations are still unsatisfactory. For patients with familial hyperlipidemia, for example, serum triglyceride levels are too high to be controlled solely by medication [2]. These patients may need dialysis therapy to remove serum lipids during childhood [3]. Some patients cannot tolerate the side effects of medication. Statins may cause rhabdomyolysis and muscle soreness, which may contribute to poor compliance. Furthermore, statins may increase the incidence of new-onset diabetes [4]. Clinically, it is not uncommon for patients receiving a statin or other lipid-lowering agent to fail to meet the therapeutic target [5].

Many studies have shown patients with chronic kidney disease (CKD) to be refractory to statin treatment, such as the 4D [6] and AURORA trials [7]. For patients undergoing hemodialysis, statin use does not improve survival rates, although statins may lower LDL levels, as reported by the Cochrane Database of Systematic Reviews [8]. On the other hand, the mortality rate of patients with CKD is higher than that of the general population [9].

For the reasons mentioned above, the need for other lipid-lowering methods is indicated, such as anion exchange resin or double filtration (DF) [10]. However, how double filtration influences hyperlipidemia and levels of persistent organic pollutants (POPs), most of which are lipophilic and hydrophobic, is still unclear. Furthermore, DM patients have a higher risk of CVD, [11] so we compared the demographic biochemical profiles between DM and non-DM patients before and after DF in order to find whether DF may influence the risk factors of CVD.

### 1.1. Persistent Organic Pollutants: Perfluorochemicals (PFCs) 

Perfluorochemicals (PFCs) are hydrophobic persistent organic pollutants. These environmental hormones are widely used in leather, textiles, and food packaging because of their stability and water-repelling properties. Once ingested through food or water, these compounds are carried by serum protein into the human body and have a long half-life [12]. PFCs form an eight-carbon perfluoroalkyl chain; all hydrogen atoms are replaced by fluorine atoms with strong bonds, contributing to PFCs’ stability against heat and ability to avoid degradation [13]. However, PFCs are toxic to the liver, immune system, and embryos [14,15]. Bioconcentration means that toxins are concentrated at much higher levels inside living creatures than in the environment, and biomagnification means that toxins produce more harmful effects on humans because humans are higher on the food chain [14].

Our study targets are perfluorooctanoic acid (PFOA) and perfluorooctane sulfonate (PFOS) because they are among the toxins with the highest concentrations in humans [16]. Compared with other toxins, PFCs’ properties and metabolism are not fully understood [17]. A large epidemiologic study showed a positive correlation between serum PFC and serum LDL [18]. According to previous studies, the half-life of PFOA is 3.8 years, and the half-life of PFOS is 5.4 years [19]. These concentration of PFCs in these patients with normal kidney function is about normal range in Taiwan, which is about 5–10 ng/mL. [12]

### 1.2. Double Filtration (DF) 

Hemodialysis (HD) is the main blood purification method worldwide. However, it can only remove small and mainly water-soluble molecules. The removal of large hydrophobic molecules by HD is not satisfactory [12,20].

Double filtration (DF) purifies blood by first separating serum and blood cells, and then removing large particles from the serum [1]. DF only removes particles larger than serum albumin after serum is separated from blood. The procedure returns the remaining filtered serum back into patients to limit blood volume loss [21]. Both filters are made of polyvinyl alcohol and have different pore sizes.

Currently, no study has evaluated the serum levels of persistent organic pollutants such as PFCs in patients with metabolic syndrome or hyperlipidemia. As a result, the serum concentrations of perfluorochemicals are still unknown, as is whether PFCs can be removed by the abovementioned methods. The effects of toxin removal on inflammation are also worthy of investigation.

## 2. Materials and Methods

### 2.1. Inclusion and Exclusion Criteria

We enrolled adults with hyperlipidemia and metabolic syndrome who poorly tolerated the available medications or were medically refractory. After informed consent, the participants had double filtration once, with serum taken before and two weeks after the therapy. We checked the serum lipid profile, inflammatory markers, and the environmental toxins PFOA and PFOS. 

We included patients aged 18 to 90 who were diagnosed as hyperlipidemic with triglycerides (TG) > 200 mg/dL or LDL > 160 mg/dL for more than three months. The exclusion criteria were age < 18 years, pregnancy or breastfeeding status, thrombocytopenia (platelet count < 100,000/mm^3^), or bleeding risk. Diabetes mellitus was defined as hemoglobin A1c (HbA1c) > 6.5%.

Also excluded from the study were patients with renal failure who had received a transplant or were undergoing hemodialysis or peritoneal dialysis, patients receiving chemotherapy for a malignancy, those who had a blood transfusion in the past two weeks or intravenous medication (such as a lipid-based nutritional supplement, propofol, dopamine, methotrexate, fluorouracil, vancomycin, prednisolone, furosemide, or cyclosporine), and those who were unwilling to sign the study agreement. A total of 45 patients were included.

Demographic and clinical data such as gender, comorbidity of diabetes mellitus, and age were obtained from medical records, and laboratory parameters were gathered prior to treatment. Blood samples were collected at a teaching hospital in northern Taiwan before and two weeks after DF. Serum complete blood count and differential count (CBC/DC) were checked, as was biochemical profile (lipid profile, liver function, renal function, uric acid, sugar, hemoglobin A1c, and high-sensitivity C-reactive protein (hsCRP). The hemogram auto-analyzer was SYSMEX XE2100 (Sysmex, Kobe, Japan). The biochemical parameters were determined by ADVIAR 1800 Chemistry System, Siemens, Germany. Our study used isotope dilution high-performance liquid chromatography coupled with mass spectrometry (IDLCMS) to quantify the level. The methods are listed in Section A.1 [18].

### 2.2. Double Filtration (DF) Protocol 

The DF session was carried out on a Plasauto with a Plasmaflo OP-08W + Cascadeflo EC30W (Asahi Kasei Medical, Tokyo, Japan). The sieving coefficients of the different filters are shown in Section A.2. The two filters differed in pore size; the first was larger, mainly for blood cell separation. The second filter with smaller pores was used to filter larger molecules in serum, such as lipoprotein [21]. Plasma rejection during the DF session varied from 5 to 15% of the treated plasma volume as a function of the secondary filter transmembrane pressure. The total time of each session was around 1–2 h. The main material for the first and second filter was polyvinyl alcohol. 

The volume of plasma treated was equal to 1.5 times the plasma volume, calculated using the Kaplan formula: PV = (0.065 × weight (kg)) × (1 − hematocrit). Blood flow was drained at 60–100 mL/min via an 18-gauge needle from one brachial vein and returned to the other brachial vein. Plasma separation was ~25% of maximum blood-flow rate. Extracorporeal anticoagulation was based on heparin, with 2000–3000 units used at the beginning of treatment and 20–40 units/kg/h used during treatment.

### 2.3. Statistical Analysis

Continuous data were expressed as the mean ± standard deviation. Statistical analyses were performed using SPSS 20.0 for Windows (web version). The paired *t*-test was employed to compare differences before and after DF. The χ^2^ test was conducted for categorical variables, and the *t*-test was used for continuous variables. Distributions of continuous variables in groups were expressed as means ± standard deviations. Non-normally distributed data were expressed as medians and interquartile ranges. A *p*-value < 0.05 was considered statistically significant (indicated with an asterisk in the figures and tables). The sample size was determined based on an effect size to detect differences in different groups. If we permitted a 5% chance of a type I error (α = 0.05), with a power of 90%, and assumed the differences among the target values before and after DF were at least equal to the standard deviation, then approximately 21 patients would be required; 45 patients were enrolled and completed the study. 

## 3. Results

As mentioned previously, 45 patients completed the study. Demographic data are listed in Table 1. Of the patients, 11 had DM, and 34 did not. No sex or age differences were noted for these two groups, except that DM patients had a much higher body weight and higher body mass index (BMI). The differences in laboratory profiles before and after DF are listed in Table 2. 

### 3.1. DF Effect on Lipid Profile

All lipid profile components were significantly lower after DF, including total cholesterol (188 to 85 mg/dL, *p* < 0.001), triglycerides (TG, 257 to 104 mg/dL, *p* < 0.001), high-density lipoprotein (HDL, 35 to 25 mg/dL, *p* < 0.001), low-density lipoprotein (LDL, 102 to 41 mg/dL, *p* < 0.001), and the marker of coronary artery disease (CAD): total cholesterol/HDL (5.77 to 3.3, *p* < 0.001, Figure 1).

### 3.2. DF Effect on Complete Blood Count (CBC)

Regarding blood cell count, white blood cell (WBC) count increased significantly (*p* = 0.033), as did the red blood cell count (RBC), Hb, hematocrit (Hct), and mean corpuscular hemoglobin (MCH) (*p* < 0.001 for all four components) For the differential count (DC), neutrophils rose significantly (53 to 66%, *p* < 0.001), while platelet decreased (*p* = 0.022) and all other types of WBCs decreased after DF (lymphocytes, *p* = 0.001; monocytes, *p* = 0.006; eosinophils, *p* < 0.001; and basophils, *p* = 0.004, Figure 2).

### 3.3. DF Effect on Liver Function, Uric Acid, Inflammation, and PFOS

In terms of inflammation, CRP decreased significantly (0.28 to 0.12, *p* < 0.001), so the inflammation likely subsided (Figure 3).

As for environmental toxins, PFOS decreased significantly (2.774 to 1.605, *p* = 0.016) and could be removed by DF. However, PFOA was not significantly decreased (3.704 to 3.717, *p* = 0.847). Serum creatinine significantly decreased (0.87 to 0.81 *p* = 0.005), implying better renal function and hence higher estimated glomerular filtration rate (eGFR, 102 to 115 mL/min, *p* = 0.027). Uric acid (6.19 to 5.97 mg/dL, *p* = 0.012) and ALT (30 to 26 U/L, *p* < 0.001) both decreased significantly, which may have resulted from an improvement in renal and liver function after toxins were removed. Over the two-week process, there were no significant changes in AST (*p* = 0.729), HbA1c (*p* = 0.929), or insulin (*p* = 0.933).

### 3.4. Comparison of DF Effect on DM and Non-DM Patients

Table 3 compares patients with and without DM. DM patients were heavier in body weight and had higher hemoglobin (Figure 4), hematocrit, serum glucose, HbA1c, insulin, ALT, and triglycerides, but significantly lower HDL and LDL compared with nondiabetic patients. A similar pattern was found after DF except for TG and T chol/HDL.

Higher TG and total cholesterol/HDL ratio were noted in the DM group before DF, but the difference became insignificant after DF (Figure 5 and Figure 6).

## 4. Discussion

WBC and neutrophil counts were significantly elevated, most likely due to the active stress induced by the puncture of vessels. DF required one puncture hole in a vein of each arm, with one site indicated for draining blood and the other site for returning the filtrated blood. DF may also concentrate the blood, activate WBC proliferation, and result in higher hematocrit (Table 2). The only blood cell type to decrease was platelets, which significantly dropped after DF because of the consumption of coagulation factor during DF [21]. 

Lower LDL may contribute to reduced inflammation via less arthrosclerosis. All lipid profile components were decreased after DF, such as total cholesterol, triglycerides (TG), high-density lipoprotein (HDL), and low-density lipoprotein (LDL). This implies that DF removed all the lipid profile components and decreased the ratio of total cholesterol/HDL, which may further lower the risk of CVD [22]. Lower LDL may also mean lower inflammation and better physical status and mood [23].

Creatinine was significantly decreased (0.87 to 0.81, *p* = 0.005). The reductions in inflammation and PFOS may contribute to better renal function and lower serum creatinine [12]. By removing environmental toxins (PFOS), DF may further improve renal and liver function, with decreased serum uric acid levels and ALT. Previous studies have shown that higher PFOS levels may induce an increase in uric acid and liver function markers [24]. Improved liver function (ALT) may have resulted from the removal of PFC [20].

The decrease in PFOS was highly correlated with the change in PFOA (*p* = 0.008). This relationship may imply that the two are removed by the same mechanism. The change in LDL, on the other hand, was not correlated with the change in PFOS, even though these two factors both decreased. Our hypothesis is that the removal of PFCs and lipids probably occurs via different mechanisms [20]. Meanwhile, higher WBCs may indicate higher blood viscosity (*p* = 0.03) and cause poor removal of PFOS due to poorer clearance [25].

For the DM versus non-DM comparison, diabetes patients were heavier in body weight, with higher hemoglobin and hematocrit, which may indicate an elevated risk of cardiovascular diseases due to an increased thrombosis risk (Table 1 and Table 3). Serum glucose, HbA1c, and insulin levels were higher in diabetic patients compared with nondiabetic patients, as expected. Higher liver function test results (ALT) and triglycerides (TG) were also noted in the diabetic group, which might relate to glycogen formation in the liver [26]. However, lower HDL and LDL also were noted in the DM group. The lipid profiles before and after DF, triglycerides, and T chol/HDL were higher in the DM group before DF, but the difference became insignificant after DF (Figure 4 and Figure 5). This finding may indicate similar CVD risks between both groups after DF [22]. Whether diabetes patients may benefit more from DF still needs more investigation.

Despite the small number of participants and short experimental period, we reached some interesting findings. Serum PFCs can be successfully removed by DF. Whether DF can remove other lipid-soluble toxins and whether PFOA can be removed under repetitive DF remain unclear. The percentage of each DF session for PFCs still needs to be verified.

Other blood purification procedures are worthy of future investigation. Through plasma exchange with the transfusion of plasma from healthy donors, disease-causing factors may be removed, such as autoantibodies in autoimmune diseases [27]. Hemoperfusion is an add-on procedure that lets blood flow into a hydrophobic filter (which normally contains active charcoal) before hemodialysis [28]. The target particles for removal are absorbable toxins. The procedure has proven to be beneficial in certain conditions, such as medication-related suicidal intoxication [29]. Further investigations of different dialysis modalities’ ability to remove environmental toxins are indicated.

## 5. Conclusions

This is the first study to show that DF may remove serum PFOS, with improved renal function, liver function, and inflammatory status.

## Figures and Tables

**Figure 1 bioengineering-10-00089-f001:**
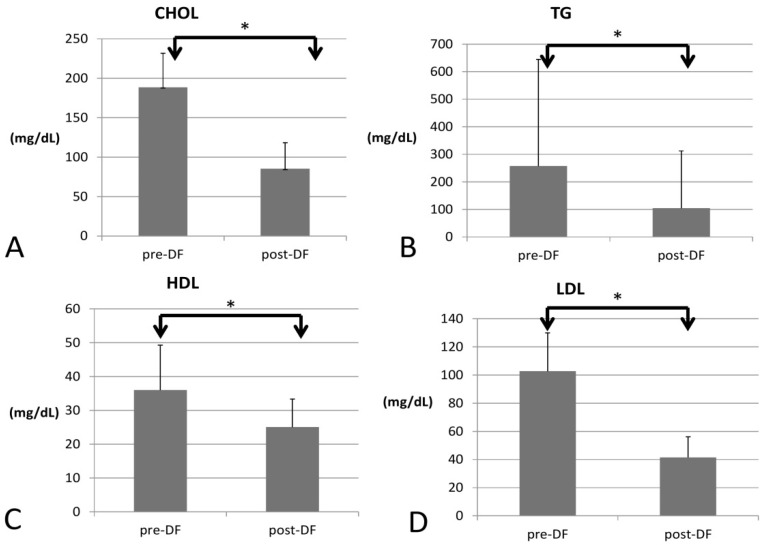
Changes in the lipid profile, including (**A**) total cholesterol (CHOL), (**B**) triglycerides (TG), (**C**) high-density lipoprotein (HDL), and (**D**) low-density lipoprotein (LDL) before and after DF (all showed significant decrease). (* for *p* < 0.05).

**Figure 2 bioengineering-10-00089-f002:**
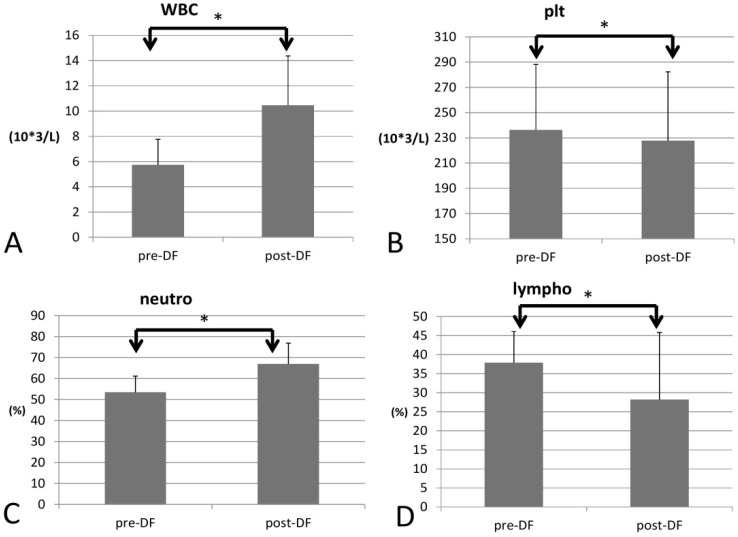
Changes in complete blood count (CBC) with increase of WBC (**A**) and decrease of platelet (**B**) and differential count (DC) with increase of neutrophil (**C**) and decrease of lymphocyte (**D**) before and after DF. (* for *p* < 0.05).

**Figure 3 bioengineering-10-00089-f003:**
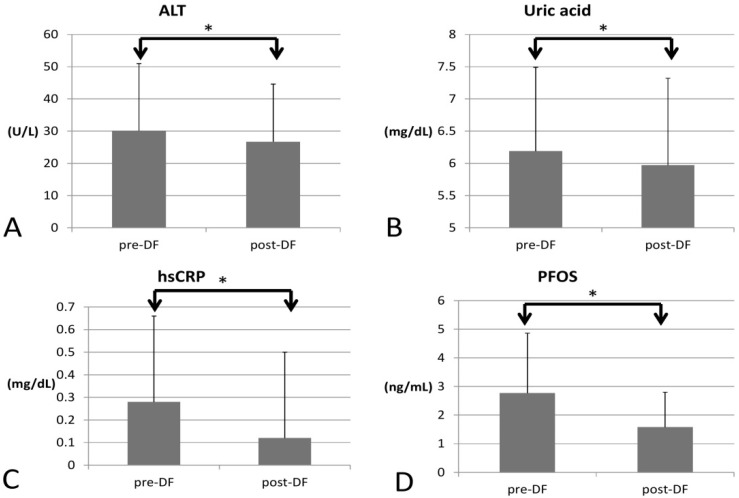
Changes in inflammatory marker, PFOS, uric acid, and liver function before and after DF (all showed significant decrease). (**A**) alanine transaminase (ALT), (**B**) Uric acid (**C**) hsCRP (**D**) perfluorooctane sulfonate (PFOS). (* for *p* < 0.05).

**Figure 4 bioengineering-10-00089-f004:**
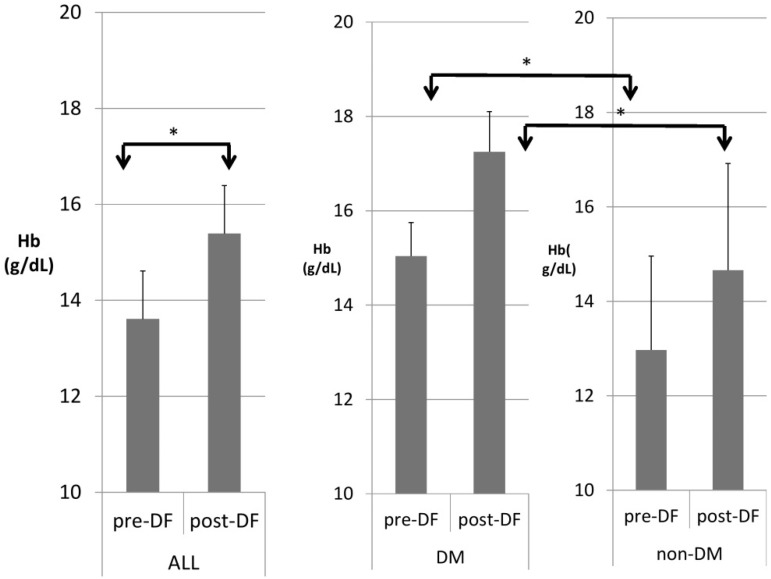
Subgroup comparison of DM and non-DM patients in hemoglobin (Hb). (* for *p* < 0.05).

**Figure 5 bioengineering-10-00089-f005:**
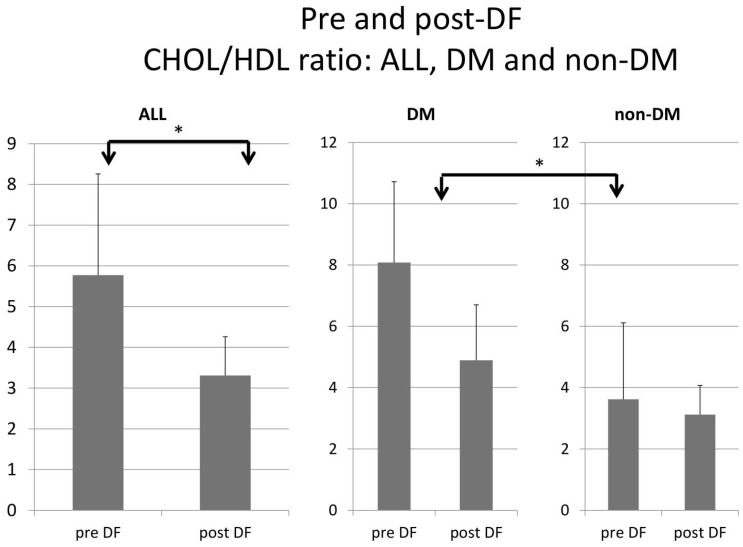
Subgroup comparison of DM and non-DM patients in total cholesterol to HDL ratio. (* for *p* < 0.05).

**Figure 6 bioengineering-10-00089-f006:**
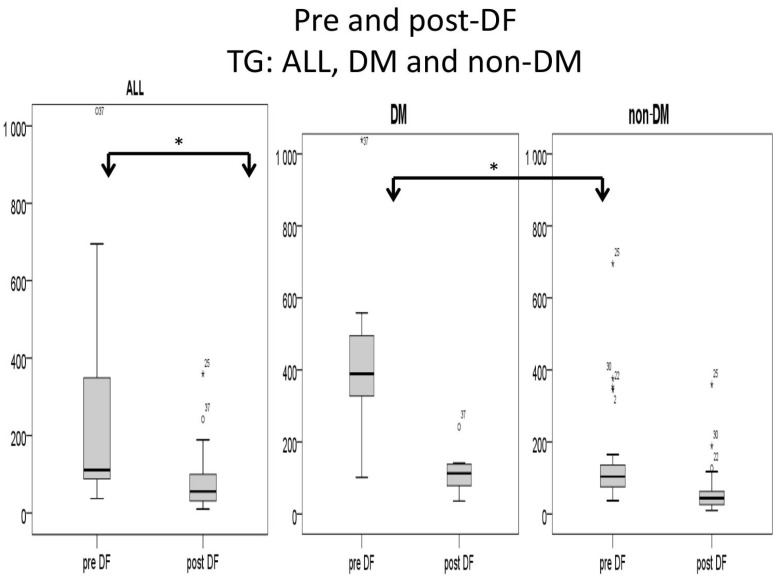
Subgroup comparison of triglyceride (TG) levels (mg/dL) for DM and non-DM patients. (* for *p* < 0.05).

**Table 1 bioengineering-10-00089-t001:** Demographic data of participants.

Characteristic	All (*n* = 45)	DM = 14	Non-DM = 31	*p*-Value
Male:Female	23:22	6:8	17:14	0.53
Age (years)	54.48 ± 10.57	57.64 ± 10.66	53.06 ± 10.38	0.182
Height (cm)	165.24 ± 7.85	167.50 ± 6.81	164.22 ± 8.18	0.199
Body weight (kg)	72.80 ± 15.94	79.64 ± 14.67	69.71 ± 15.75	0.05
Body mass index (BMI)	26.44 ± 4.29	28.20 ± 3.14	25.65 ± 4.54	0.037 *

DM, diabetes mellitus; * *p* < 0.05.

**Table 2 bioengineering-10-00089-t002:** Biochemical profile before and after DF, and linear regression of changed values with PFOS.

*n* = 45 Male = 23	before DF (mean ± SD)	after DF (Mean ± SD)	*p*-Value	Linear Regression of Δ Value to Δ PFOS (R/p)
WBC 10^3^/L	5.73 ± 2.03	10.46 ± 3.91	<0.001 *	−0.289 (0.033) *
RBC 10^6^/L	4.58 ± 0.67	5.18 ± 0.77	<0.001 *	−1.286 (0.606)
Hb (g/dL)	13.61 ± 1.96	15.39 ± 2.26	<0.001 *	−0.690 (0.476)
HCT (%)	41.76 ± 5.82	46.02 ± 6.07	<0.001 *	0.064 (0.858)
MCV (fl)	91.93 ± 6.03	90.31 ± 5.13	0.005 *	0.363 (0.331)
MCH (pg)	29.81 ± 1.66	29.95 ± 1.59	0.083	−2.680 (0.208)
Platelets 10^3^/L	236.30 ± 51.79	227.78 ± 54.55	0.022 *	−0.025 (0.104)
Neutrophils (%)	53.40 ± 7.77	66.93 ± 9.92	<0.001 *	−0.015 (0.823)
Lymphocytes (%)	37.87 ± 8.19	28.20 ± 17.63	0.001 *	0.005 (0.939)
Monocytes (%)	5.56 ± 1.79	4.60 ± 1.28	0.006 *	0.048 (0.874)
Eosinophils (%)	2.51 ± 1.45	2.12 ± 1.28	<0.001 *	0.057 (0.879)
Basophils (%)	0.65 ± 0.35	0.52 ± 0.27	0.004 *	0.251 (0.869)
Cr (mg/dL)	0.87 ± 0.70	0.81 ± 0.71	0.005 *	−7.152 (0.067)
eGFR (mL/min)	102.45 ± 30.86	115.12 ± 45.05	0.033 *	0.036 (0.415)
Uric acid (mg/dL)	6.19 ± 1.30	5.97 ± 1.35	0.032 *	2.277 (0.482)
Glucose (mg/dL)	118.63 ± 53.83	137.67 ± 101.70	0.190	0.003 (0.973)
HbA1c (%)	6.25 ± 1.52	6.25 ± 1.55	1	9.047 (0.302)
Insulin (mU/L)	13.63 ± 9.24	13.50 ± 11.73	0.975	0.087 (0.193)
AST (U/L)	26.00 ± 15.56	25.70 ± 15.23	0.755	−0.041 (0.772)
ALT (U/L)	30.14 ± 20.81	26.70 ± 17.91	<0.001 *	−0.126 (0.280)
CHOL (mg/dL)	188.60 ± 43.12	85.23 ± 33.01	<0.001 *	0.025 (0.098)
TG (mg/dL) median, IQR for TG	257.5 ± 387.47 111 (87.25–349.75) ^+^	104.25 ± 208.06 56 (31–100) ^+^	<0.001 *	0.006 (0.307)
HDL (mg/dL)	35.96 ± 13.29	25.03 ± 8.29	<0.001 *	−0.064 (0.224)
LDL (mg/dL)	102.67 ± 27.28	41.48 ± 14.62	<0.001 *	0.036 (0.175)
CHOL/HDL ratio	5.77 ± 2.49	3.31 ± 0.95	<0.001 *	0.479 (0.121)
hsCRP (mg/dL)	0.28 ± 0.38	0.12 ± 0.18	<0.001 *	4.301 (0.419)
PFOA (ng/mL)	3.70 ± 2.54	3.87 ± 2.78	0.847	18.951 (0.008) *
PFOS (ng/mL)	2.77 ± 2.09	1.58 ± 1.22	0.017 *	-

DF, double filtration; WBC, white blood cell count; RBC, red blood cell count; Hb, hemoglobin; HCT, hematocrit; MCV, mean corpuscular volume; MCH, mean corpuscular hemoglobin; Cr, creatinine; eGFR, estimated glomerular filtration rate; HbA1c, hemoglobin A1c; AST, aspartate transaminase; ALT, alanine aminotransferase; CHOL, cholesterol; TG, triglyceride; HDL, high-density lipoprotein cholesterol; LDL, low-density lipoprotein cholesterol; Tcho/HDL ratio, total cholesterol to HDL; hs CRP, high sensitive C reactive protein; PFOA, perfluorooctanoic acid; PFOS, perfluorooctane sulfonate; * *p* < 0.05; ^+^ IQR: interquartile range.

**Table 3 bioengineering-10-00089-t003:** Comparison of DM (n = 11) vs. non-DM (n = 31) patients before and after DF.

*n* = 45 (Mean ± SD)	before DF: DM pt	before DF: Non-DM pt	*p*-Value	after DF: DM pt	after DF: Non-DM pt	*p*-Value
WBC 10^3^/L	5.82 ± 1.39	5.71 ± 2.23	0.888	10.86 ± 1.43	10.34 ± 4.56	0.713
RBC 10^6^/L	4.92 ± 0.17	4.41 ± 0.71	0.001 *	5.63 ± 0.19	4.94 ± 0.83	0.010
Hb (g/dL)	15.04 ± 0.71	12.97 ± 1.99	<0.001 *	17.25 ± 0.85	14.66 ± 2.26	0.001
HCT (%)	46.53 ± 2.94	39.88 ± 5.67	<0.001 *	50.76 ± 2.29	44.49 ± 6.27	<0.001 *
MCV (fl)	94.48 ± 5.40	90.93 ± 6.09	0.096	90.12 ± 2.72	90.39 ± 5.81	0.887
MCH (pg)	30.58 ± 1.17	29.50 ± 1.74	0.065	30.61 ± 0.96	29.71 ± 1.74	0.111
Platelets 10^3^/L	217.36 ± 44.46	245.06 ± 54.17	0.136	211.54 ± 56.55	233.73 ± 54.47	0.260
Neutrophils (%)	55.93 ± 5.64	52.62 ± 8.33	0.231	67.15 ± 3.98	66.85 ± 11.55	0.901
Lymphocytes (%)	35.61 ± 6.06	28.50 ± 8.79	0.311	26.39 ± 3.27	25.63 ± 10.20	0.722
Monocytes (%)	5.98 ± 1. 93	5.37 ± 1.76	0.341	4.15 ± 6. 83	4.76 ± 1.42	0.187
Eosinophils (%)	1.90 ± 1.12	2.72 ± 1.50	0.105	0.50 ± 0.20	0.52 ± 0.29	0.238
Basophils (%)	0.56 ± 0.39	0.70 ± 0.34	0.279	0.56 ± 0.39	0.70 ± 0.34	0.782
Cr (mg/dL)	0.85 ± 0.20	0.86 ± 0.82	0.952	0.83 ± 0.17	0.79 ± 0.84	0.894
eGFR (mL/min)	100.33 ± 20.71	103.39 ± 35.72	0.824	104.00 ± 27.41	119.10 ± 52.14	0.426
Uric acid (mg/dL)	6.11 ± 0.86	6.23 ± 1.58	0.821	6.19 ± 0.85	5.92 ± 1.62	0.821
Glucose (mg/dL)	191.00 ± 43.28	90.61 ± 23.65	<0.001 *	180.58 ± 40.85	121.06 ± 114.85	0.049 *
HbA1c (%)	8.67 ± 1.02	5.43 ± 0.39	<0.001 *	8.62 ± 1.05	5.43 ± 0.44	<0.001 *
Insulin (mU/L)	19.50 ± 10.35	11.47 ± 8.16	0.013 *	15.59 ± 7.24	12.29 ± 12.24	0.48 *
AST (U/L)	22.00 ± 5.65	26.88 ± 17.14	0.710	17.00 ± 5.65	26.66 ± 16.71	0.568
ALT (U/L)	44.83 ± 16.35	24.45 ± 20.06	0.003	39.91 ± 14.00	21.58 ± 17.05	0.002
CHOL (mg/dL)	206.63 ± 57.73	183.64 ± 35.51	0.128	85.75 ± 48.72	85.44 ± 25.09	0.979
TG (mg/dL) ^+^	389 (306–558)	104 (74–141)	0.048	106.5 (76.75–140.25)	44 (22–64)	0.186
HDL (mg/dL)	24.63 ± 5.57	40.45 ± 12.91	<0.001 *	20.16 ± 5.82	27.24± 8.42	0.012
LDL (mg/dL)	88.18 ± 20.11	108.64 ± 27.79	<0.001 *	33.50 ± 9.47	45.13 ± 15.26	0.019
CHOL/HDL ratio	8.08 ± 2.64	4.89 ± 1.81	0.002	3.62 ± 0.61	3.12 ± 1.05	0.242
hsCRP (mg/dL)	0.34 ± 0.11	0.25 ± 0.44	0.491	0.12 ± 0.068	0.11 ± 0.21	0.767
PFOA (ng/mL)	2.20 ± 1.93	2.96 ± 2.75	0.623	2.18 ± 0.66	4.22 ± 2.96	0.276
PFOS (ng/mL)	1.93 ± 2.46	2.07 ± 2.03	0.917	1.32 ± 0.93	1.69 ± 1.27	0.654

DF, double filtration; DM, diabetes mellitus; pt, patients; WBC, white blood cell count; RBC, red blood cell count; Hb, hemoglobin; HCT, hematocrit; MCV, mean corpuscular volume; MCH, mean corpuscular hemoglobin; Cr, creatinine; eGFR, estimated glomerular filtration rate; HbA1c, hemoglobin A1c; AST, aspartate transaminase; ALT, alanine aminotransferase; CHOL, cholesterol; TG, triglycerides; HDL, high-density lipoprotein cholesterol; LDL, low-density lipoprotein cholesterol; Tcho/HDL ratio, total cholesterol to HDL; hsCRP, high sensitive C reactive protein; PFOA, perfluorooctanoic acid; PFOS, perfluorooctane sulfonate; * *p* < 0.05, ^+^ IQR: interquartile range.

## Data Availability

Not applicable.

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
