# Peer review of "Double Filtration Plasmapheresis with Polyvinyl Alcohol-Based Membrane Lowers Serum Inflammation and Toxins in Patients with Hyperlipidemia"

_bioengineering, 2023, doi:10.3390/bioengineering10010089_

Round 1

Reviewer 1 Report

The manuscript studied the effect of Double filtration plasmapheresis on lipid profile, ALT As marker of liver function, complete blood count and differential count, inflammation and toxins mainly perfluorooctane sulfonate in patients with hyperlipidemia. The authors demonstarated that Double filtration plasmapheresis  remove serum lipids and perfluorooctane sulfonate and improve liver function and inflammation.

My comments

-The mansucripts is confusing, the authors divided the hyperlipidemic patients into diabetic and non diabetic, while the aim of the work and conclusion was nothing about diabetes, please explain

-The abstract needs clarification: Double filtration (DF) is sometimes abbreviated as DFPP. Also, The total cholesterol/HDL ratio and triglyceride were significant higher in the DM group:  DM was first mentioned with no clarifications in methods part 

-In introduction: the first sentence needs reference.  line 52: "For patients who can’t tolerate the side effect of medication": this sentence is not complete

-The authors used polyvinyl alcohol-based membranes: are there any other types of filters used?

- Were perfluorochemicals  high in all selected patients? Were the patient with normal or low level excluded? this point was not clarified

-Line 101 "We checked and 101 the serum lipid profile": something is missed 

-Methods for measuring biochemical and inflamamtory markers should be indicated in the manuscript, kits used and catalog numbers

-Check numbers through results for appropriate writing of numbers with decimals

-The figure legends should eb improved: abbreviations should be verified and if data are parametric or non-parametric. How data were presented mean and standard deviations or using medians?. 

-Why ALt was only used for liver function?

-The results section should include subtitles

-Figure 4 did not show CBC, please revise

-Figure 5 is not clear, the quality needs improvement

-Discussion needs to be re-written: no need to repeat the results nor recite tables and figures, instead refer to previous research with similar findings 

Author Response

Reviewer 1

The manuscript studied the effect of Double filtration plasmapheresis on lipid profile, ALT As marker of liver function, complete blood count and differential count, inflammation and toxins mainly perfluorooctane sulfonate in patients with hyperlipidemia. The authors demonstarated that Double filtration plasmapheresis  remove serum lipids and perfluorooctane sulfonate and improve liver function and inflammation.

My comments

1.-The mansucripts is confusing, the authors divided the hyperlipidemic patients into diabetic and non diabetic, while the aim of the work and conclusion was nothing about diabetes, please explain

2.-The abstract needs clarification: Double filtration (DF) is sometimes abbreviated as DFPP. Also, The total cholesterol/HDL ratio and triglyceride were significant higher in the DM group:  DM was first mentioned with no clarifications in methods part

3.-In introduction: the first sentence needs reference.  line 52: "For patients who can’t tolerate the side effect of medication": this sentence is not complete

4.-The authors used polyvinyl alcohol-based membranes: are there any other types of filters used?

5.- Were perfluorochemicals  high in all selected patients? Were the patient with normal or low level excluded? this point was not clarified

6.-Line 101 "We checked and the serum lipid profile": something is missed

7.-Methods for measuring biochemical and inflammatory markers should be indicated in the manuscript, kits used and catalog numbers

8.-Check numbers through results for appropriate writing of numbers with decimals

9.-The figure legends should be improved: abbreviations should be verified and if data are parametric or non-parametric. How data were presented mean and standard deviations or using medians?.

10.-Why ALt was only used for liver function?

11.-The results section should include subtitles

12.-Figure 4 did not show CBC, please revise

13.-Figure 5 is not clear, the quality needs improvement

14.-Discussion needs to be re-written: no need to repeat the results nor recite tables and figures, instead refer to previous research with similar findings

Answers:

  1. Diabetes patients are prone to coronary artery disease (CAD) and CAD is highly related to serum lipid. By lowering serum lipid, especially LDL, the risk of CAD may be lower. We found the CAD risk index of total cholesterol/HDL became similar “after” double filtration (DF) in DM and non DM group, which may be indicate the CAD risk became similar in these two groups.
  2. Double filtration plasmapheresis (DFPP) is the same as Double filtration (DF). I will unify the terms. DM is defined as HbA1c higher than 6.5%.
  3. The introduction “For patients who can’t tolerate the side effect of medication" is corrected to “Some patients can’t tolerate the side effect of medication.[1]” Thanks for your kindly reminding.
  4. In our study, double filtration filters are only used with ethylene-vinyl alcohol based membranes. There are also other double filtration filters membranes made of polyethersulfone. [2]There are other kinds of membranes, such as polysuflone or cellulose used in hemopurification, but designed for other purpose, such as hemodialysis or hemofiltration.
  5. These concentration of PFC in these patients with normal kidney function is about normal range in Taiwan, which is about 5-10 ng/ml.[3]All patients’ data are included in the analysis. We will add this in the discussion.
  6. “We checked and the serum lipid profile”we corrected it and we appreciated for your kindly reminding.
  7. The hemogram auto-analyzer was SYSMEX XE2100 (Sysmex, Kobe, Japan). The iochemical parameters are determined by ADVIAR 1800 Chemistry System, Siemens, Germany.
  8. The numbers in results are checked and restricted to 2 digits below decimal, such as “ PFOS decreased significantly (2.77 to 1.58, p=0.017) and can be removed by DF
  9. The figures will be improved in quality. Also the data are all parametric except for TG. So the TG before and after DF is presented again in figure 5 with box and whisker plot.
  10. ALT and AST are used to check liver function, but only ALT showed significantly decreased after DF. The reason may be related to the relation of TG with diabetes.[4]
  11. The subtitles will be added to the results. Thanks for your excellent advice.
  12. We added another new Figure 4 showing the change of Hb before and after DF in all, DM and non-DM patients.
  13. We will improved the quality of all figures
  14. Discussion will be rewritten and English polished.

Reference

  1.            Ward, N.C.; Watts, G.F.; Eckel, R.H. Statin Toxicity. Circulation research 2019124, 328-350, doi:10.1161/circresaha.118.312782.
  2.             Krieter, D.H.; Jeyaseelan, J.; Rüth, M.; Lemke, H.D.; Wanner, C.; Drechsler, C. Clinical hemocompatibility of double-filtration lipoprotein apheresis comparing polyethersulfone and ethylene-vinyl alcohol copolymer membranes. Artificial organs 202145, 1104-1113, doi:10.1111/aor.13944.
  3.             Liu, W.S.; Lai, Y.T.; Chan, H.L.; Li, S.Y.; Lin, C.C.; Liu, C.K.; Tsou, H.H.; Liu, T.Y. Associations between perfluorinated chemicals and serum biochemical markers and performance status in uremic patients under hemodialysis. PLoS One 201813, e0200271, doi:10.1371/journal.pone.0200271.
  4.             Song, S.; Zhang, Y.; Qiao, X.; Duo, Y.; Xu, J.; Peng, Z.; Zhang, J.; Chen, Y.; Nie, X.; Sun, Q.; et al. ALT/AST as an Independent Risk Factor of Gestational Diabetes Mellitus Compared with TG/HDL-C. International journal of general medicine 202215, 115-121, doi:10.2147/ijgm.s332946.

Reviewer 2 Report

The manuscript entitled ‘Double filtration plasmapheresis with polyvinyl alcohol-based 2 membrane lowers serum inflammation and toxins in patients 3 with hyperlipidemia’ by Liu etal.

Describes the use of PVA based membrane to remove toxins from serum of patients with hyperlipidemia.

Overall, the study may be interesting, but the way the manuscript is written and presented is not acceptable. In addition to significant issues with comprehension, grammar, spelling mistakes, not defining the acronyms, and spelling mistakes, the introduction itself is misleading. This appear that the study aims to look at three different application of this membrane, but this is not the case.

The specification of the membrane used is missing so this is unclear, what are we looking at in terms of results. The study also contains several blurred images, and axis legend are missing for figure 1,2,3,4 and 5.

The study cannot be defined as a manuscript and is hardly a report. I would not recommend publishing this in this journal.

Author Response

Reviewer 2: The manuscript entitled ‘Double filtration plasmapheresis with polyvinyl alcohol-based membrane lowers serum inflammation and toxins in patients with hyperlipidemia’ by Liu etal. Describes the use of PVA based membrane to remove toxins from serum of patients with hyperlipidemia.

  1. Overall, the study may be interesting, but the way the manuscript is written and presented is not acceptable.  In addition to significant issues with comprehension, grammar, spelling mistakes, not defining the acronyms, and spelling mistakes, the introduction itself is misleading. This appears that the study aims to look at three different application of this membrane, but this is not the case. The specification of the membrane used is missing so this is unclear, what are we looking at in terms of results.
  2. The study also contains several blurred images, and axis legends are missing for figure 1,2,3,4 and 5. The study cannot be defined as a manuscript and is hardly a report. I would not recommend publishing this in this journal.

Answers:

  1. I will do English polishing and rewrite the introduction to correct the grammar errors. I am sorry for the quality of English at the initial manuscript.

We are testing the clinical effect of double filtration on serum lipid and inflammation for patients with hyperlipidemia. We found the biochemical benefit can be explained by the removal of lipophilic toxin with this polyvinyl alcohol membrane.

Our manuscript was originally sent to the special issue of “Advanced Polymer Membranes for Adsorption and Separation Applications” in “polymers”, so it would be confusing why we mentioned polymer based membrane in the first place.

  1. The quality of images will be improved, and the axis legends will be added as well.

I hope our rewriting will be suitable for bioengineering.

Round 2

Reviewer 1 Report

The authors addressed my comments and the manuscript has been improved